

# **Modeling the Inhibition Effect of Straw Checkerboard**
# **Barriers on Wind-blown Sand**
**Haojie Huang[1,2,3]**
[1]School of Energy and Power Engineering, University of Shanghai for Science and Technology,
Shanghai, PR China
[2]MOE Engineering Research Center of Desertification and Blown-sand Control, Beijing Normal
University, Beijing, PR China
[3]College of Mechanics and Materials, Hohai University, Nanjing, Jiangsu 211000, PR China
**Correspondence:** H.J. Huang (hjhuang@usst.edu.cn)
## **Abstract**
Straw checkerboard barriers (SCBs) are usually laid to prevent or delay the process of
desertification caused by aeolian sand erosion in arid and semi-arid regions.
Understanding the impact of SCBs and its laying length on aeolian sand erosion is of
great significance to reduce the damage and the laying costs. In this study, a
three-dimensional wind-blown sand model in presence of SCBs was established by
introducing the splash process and equivalent sand barriers into a large-eddy
simulation airflow. From this model, the inhibition effect of SCBs on wind-blown
sand was studied qualitatively, and the sensitivity of aeolian sand erosion to the laying
length was investigated. The results showed that the wind speed in the SCBs area
decreases oscillatively along the flow direction. Moreover, the longer the laying
lengths, the lower the wind speed in the stable stage behind SCBs, and the lower the



sand transport rate. We further found that the concentration of sand particles near the
side of SCBs is higher than that in its central region, which is qualitatively consistent
with the previous research. Our results also indicated that whether the wind speed will
decrease below the impact threshold or the fluid threshold is the key factor affecting
whether sand particles can penetrate the SCBs and form stable wind-blown sand
behind the SCBs under the same conditions. Our research can provide theoretical
support for the minimum laying length of SCBs in anti-desertification projects.
**1. Introduction**

In arid and semi-arid areas, aeolian sand erosion is becoming more and more

serious. How to prevent or delay the process of desertification is a major challenge for
all over the world, especially in the transitional areas between desert and oasis. At
present, shelterbelt (Wang et al., 2010), sand fences (Bitog et al., 2009; Hatanaka and
Hotta, 1997; Li and Sherman (2015); Lima et al., 2017; Pye and Tsoar, 2008; Wilson,
2004), wind-break walls (Bouvet et al., 2006; Santiago et al., 2007), hole plate-type
sand barriers (Chen et al., 2019) and straw checkerboard sand barriers (Bo et al., 2015;
Huang et al., 2013; Wang and Zheng, 2002; Xu et al., 2018) are the main techniques
to prevent the process of desertification. Among these techniques, straw checkerboard
barriers (SCBs) are the most common used in the anti-desertification projects because
of their advantages of easy to obtain and relatively cheap (Zheng, 2009). Laying
SCBs could play an important role in the ecological restoration of sandy land
ecosystems (Zhang et al., 2018) and vegetation restoration. Some research showed
that the SCBs can effectively reduce the surface wind speed (Qu et al., 2007), increase





the surface roughness (Zhang et al., 2016), weaken the sand transport rate (Bo et al.,
2015), change the distribution of aeolian sandy soil particles and soil organic carbon
(Dai et al., 2019), thus protecting the survival of the vegetation and achieving
sustainable development of oasis and ecological environment.
In recent decades, SCBs have been widely used in northwest of China, which is
seriously damaged by aeolian sand erosion. For example, the SCBs have been laid on
the sides of the roadbed along the railways such as Baotou-Lanzhou Railway,
Wuda-Jilantai Railway (Wang, 1996), Gantang-Wuwei Railway (Yang, 1995),
Lanzhou-Xinjiang Railway (Binwen et al., 1998; Cheng et al., 2016), Qinghai-Tibet
Railway (Cheng and Xue, 2014; Zhang et al., 2010), as well as the windy sand area
beside the desert roads such as Taklamakan Desert Highway (Li et al., 2006; Qu et al.,
2007), Tarim Desert Highway (Xu et al., 1998) and MinQin Desert Highway. In
addition, SCBs are adopted by some countries that are also affected by aeolian sand
erosion, such as Ghana, Egypt, and Iran (Zheng, 2009). Although the SCBs have been
widely used, its design size and laying methods are mainly determined by practical
experience or repeated tests. For example, for the sand fence which has the similar
effect to the SCB, Li and Sherman (2015) combined experimental and field data to
conclude that the optimal design of sand fence is closely related to its aerodynamics
and morphodynamics. The effect of sand fences with different porosity, spacing and
height on the wind field is significant (Lima et al., 2017; Lima et al., 2020). However,
the complexity of the flow field around the SCBs and the movement of sand particles,
as well as the coupling of particles and flow field, which makes this problem more



difficult. Therefore, it is necessary to study the characteristics of turbulence inside and
behind the SCBs as well as the influence mechanism of the laying length on wind
speed and erosion.

Wang and Zheng (2002) proposed a single-row ideally uniformly distributed

vortex model to simplify the flow field of the wind-blown sand. Based on their model,
the corresponding relationship between the side length and the height of a single SCB
was analyzed. Their theoretical results are similar to the size of the SCBs which laid
in the Tarim Desert Highway (1 m in side length and 15-20 cm in height). Qiu et al.
(2004) pointed out that since the concentration of wind-blown sand below 10 cm near
the surface is relatively high, the height of the SCBs should be designed as 10-20cm
thus to effectively prevent aeolian sand erosion. The experimental results of Zhang et
al. (2018) indicated that the SCB has the best protective effect when its side length is
1 m. These works are of great help to the design of a single SCB. Based on these
empirical sizes of the SCBs, researchers tried to analyze the effect of SCBs on the
flow field and particles from the perspective of turbulence. Huang et al. (2013) used
two-dimensional large eddy simulation and discrete particle tracking methods to
simulate the wind-blown sand movement inside the simplified two-dimensional SCBs.
The effect of SCBs on surface wind speed was analyzed. They found that sand
particles could be aggregated at the inner walls of the SCBs due to the influence of the
vortex or the backflow. And then a v-shaped sand trough was formed, which is similar
to the actual situation. Bo et al. (2015) equated the SCBs to the source term of the
standard $k$-$\varepsilon$ turbulence model, and analyzed the influence of SCBs on the wind speed



profile in two-dimensional flow field without sand particles. They divided the
streamwise velocity profile in flow field containing SCBs into three different
log-linear functions approximately, and obtained the relationship between them and
friction wind speeds. Although these two-dimensional models can reflect the effect of
the SCBs on the flow field to some extent, they are far from the real turbulence.
Moreover, since the actual three-dimensional SCB is simplified into two-dimensional
plane with only streamwise direction and vertical direction. And the impact of this
simplification is uncertain. For this reason, Xu et al. (2018) simulated the wind-blown
sand movement on the SCBs surface under three-dimensional flow field with
OpenFOAM, and mainly analyzed the influence of the flow field inside the SCBs on
the movement of sand particles. They concluded that the wind vortex is the main
cause of internal morphology of the straw checkerboard. They found that the vortex
will drive particles inside the SCBs move towards the front and side walls, making the
erosional form in SCB cells become low in the middle and high near all sides.
However, the SCBs are completely equivalent to the solid as the bottom boundary
condition in their model. As a non-solid material, the SCBs can be penetrated by the
wind in practice. It only weakens the wind speed thus not equivalent to a solid. For
example, Dupont et al. (2014) equated the surface vegetation to a resistance force
through the resistance coefficient and leaf area coefficient, that is, the wind will be
resisted as it passes through these equivalent regions.

In order to reasonably introduce the SCBs and consider the coupling among

turbulence, SCBs and surface splash process, the development of three-dimensional



model is required. In this paper, three-dimensional numerical coupled model of
wind-blown sand in presence of SCBs was carried out to study the inhibition effect of
the laying length on aeolian sand erosion. The large eddy simulation approach was
used to simulate the clean air flow with the saltation process was considered.
Furthermore, we added a volume drag force into the Navier-Stokes equations by using
the drag source method to realize the coupling between the SCBs and the wind-blown
sand movement. Section 2 and Section 3 gave the three-dimensional numerical
coupled model and its validation, respectively. In Section 4, the effects of the SCBs'
laying length on clean air flow and sand-laden flow under different friction wind
speeds were studied. Finally, Section 5 was a summary of the main conclusions.

## 122   2. Models

The Advanced Regional Prediction System (ARPS) has been widely used to
simulate turbulent boundary-layer particle-laden flow, such as: wind-blown sand
(Dupont et al., 2013; Huang, 2020), wind-blown snow (Huang and Wang, 2016; Li et
al., 2018). The standard version of the program is described in the ARPS User's
Manual (Xue et al., 1995) and its validation cases are referred to Xue et al. (2000) and
Xue et al. (2001). For this study, some suitable models were added in order to
simulate turbulent boundary-layer flow in presence of SCBs with saltating sand
particles. A detailed description of these modifications is shown in the following
subsections.

### 132   2.1 Turbulent boundary-layer flow





Basic flow fields in our numerical simulation are established on the basis of the
ARPS (version 5.3.4). And the filtered continuity and momentum equations including
viscous drag force terms of sand particles as well as SCBs are shown as follows
(Dupont et al., 2013; Vinkovic et al., 2006):
$$\frac{\partial \tilde{u}_i}{\partial t} + \tilde{u}_j \frac{\partial \tilde{u}_i}{\partial x_j} = -\frac{1}{\overline{\rho}_f}\frac{\partial}{\partial x_i}(\tilde{p}-v\frac{\partial \overline{\rho}_f \tilde{u}_j}{\partial x_j}) - \frac{\partial \tau_{ij}}{\partial x_j} - \delta_{i3}g(\frac{\tilde{\theta}}{\overline{\theta}}-\frac{c_p}{c_v}\frac{\tilde{p}}{\overline{p}}) + \frac{F_i}{\overline{\rho}_f}, \qquad (1)$$

where, $i$ = 1, 2 and 3 correspond to the streamwise, spanwise and wall-normal
directions (i.e., $x_1 = x$, $x_2 = y$, $x_3 = z$, $u_1 = u$, $u_2 = v$, $u_3 = w$), respectively; $\tilde{u}_i$, $\tilde{p}$ and
$\tilde{\theta}$ represent the filtered wind speed, pressure and potential temperature, respectively; $v$
is a damping coefficient of the attenuate acoustic waves; $\rho_f$ is the air density; $g$ is the
acceleration of gravity; $F_i$ is the feedback force of sand particles and SCBs; $\delta_{ij} = 1$ if
$i = j$ , otherwise $\delta_{ij} = 0$ ; $\tau_{ij} = \widetilde{u_i u_j} - \tilde{u}_i \tilde{u}_j$ are the SGS (sub-grid-scale) stresses
(Smagorinsky, 1963); $c_p$ and $c_v$ are the specific heat of air at constant pressure and
volume, respectively.
In order to solve above equations, the SGS stresses can be closed as follows:
$$\tau_{ij} - \frac{1}{3}\tau_{kk}\delta_{ij} = -(C_{sgs}\Delta)^2 \frac{1}{\sqrt{2}}\left|\frac{\partial \tilde{u}_i}{\partial x_j}+\frac{\partial \tilde{u}_j}{\partial x_i}\right|(\frac{\partial \tilde{u}_i}{\partial x_j}+\frac{\partial \tilde{u}_j}{\partial x_i}), \qquad (2)$$

where $\Delta$ is the grid scale; $C_{sgs}$ depends on the Germano subgrid-scale closure
method (Germano et al., 1991).
For the governing equations mentioned above, periodic boundary conditions are
applied for spanwise direction. The upper and lower boundaries are set as a stress-free
condition and a rigid ground condition, respectively. The outlet boundary is used as an



open radiation condition in this paper. The inlet boundary is a given logarithmic
profile:
$$\tilde{u}(0, y, z) = (\frac{u_*}{\kappa}) \ln(\frac{z}{z_0}). \tag{3}$$
Here $k$=0.41 is von Kármán constant; $z_0$=D/30 is the aerodynamic surface roughness
(Kok et al., 2012); $u_*$ is the friction speed of inflow. Additionally, the simulation is
driven by a constant flow corresponding to the given logarithmic wind profile. In
order to accelerate the development of boundary layer flow, LWS method (Lund et al.,
1998) are applied to the inlet condition and the recycling plane at $x_{ref}$ =5m
($x_{ref}$/$Lx$=12.5% (Inoue and Pullin, 2011), see Fig. 1). The specific method is to
re-assign the calculated mean velocity and fluctuation at the recycling plane to the
inlet at each fluid time step. There is a similar application in the paper of Xu et al.

(2018).

**2.2 Movement of sand particles**
Saltating particles are moved by the drag force, gravity, electric field force,
Magnus force, Saffman force and so on (Murphy and Hooshiari, 1982). In our model,
the drag force and gravity are considered, ignoring other minor factors (Kok et al.,
2012; Zou et al., 2007). We employ the Lagrangian point-particle method to describe
particle motions, and the equations of particles with different sizes in three directions
can be expressed as
$$m_p \frac{d^2 x}{dt^2} = \frac{C_D \pi D^2 \rho_f}{8} (\tilde{u} - \frac{dx}{dt})^2 + F_{nx} + F_{sx}, \tag{4}$$



$$m_p \frac{d^2 y}{dt^2} = \frac{C_D \pi D^2 \rho_f}{8}(\tilde{v}-\frac{dy}{dt})^2 + F_{ny} + F_{sy},$$  (5)
$$m_p \frac{d^2 z}{dt^2} = -\frac{\pi g \rho_p D^3}{6} + \frac{C_D \pi D^2 \rho_f}{8}(\tilde{w}-\frac{dz}{dt})^2 + F_{nz} + F_{sz},$$  (6)
where $m_p$ is the mass of sand particles; $C_D$ is the drag coefficient of sand particles
(Cheng, 1997). The particle Reynolds number can be expressed as
$$\mathrm{Re}_p = (V_f \rho_p D / \mu)[(\tilde{u}-dx/dt)^2 + (\tilde{v}-dy/dt)^2 + (\tilde{w}-dz/dt)^2]^{1/2}.$$  (7)
$\rho_p$ and $\rho_f$ are the density of sand particles and air, respectively; $D$ is the diameter of
sand particles; $V_f = 1 - \sum\limits_{k=1}^{k=n} V_P / \Delta V$ is the bulk fraction which is the total sand volumes
within grid to the bulk of unit grid; $\Delta V$ is the bulk of unit grid; $\mu$ is the kinetic
viscosity coefficient of air; $F_{nx}$, $F_{sx}$, $F_{ny}$, $F_{sy}$, $F_{nz}$ and $F_{sz}$ are the normal and tangential
force of contact in three directions.
**2.3 Particle collision**
The collision process in the air among the ejection particles is focused in
previous models (Carneiro et al., 2013; Huang et al., 2007). In this paper, the
"spring-damping" model is used to calculate the contact force when particles collide
in the air. And the contact force can be described as follows (Huang et al., 2017):
The normal force of contact is
$$\vec{F}_{n,ij} = \begin{cases} -k_n \zeta_{n,ij} \vec{n}_{ij} - d_n \vec{v}_{n,ij} &, \quad \zeta = \left| \mathrm{R}_i + R_j - \vec{r}_{ij} \right| \\ 0 &, \quad \zeta < 0 \end{cases}.$$  (8)
Where, $k_n = 2 \times 10^6$ is the normal stiffness coefficient; $\zeta$ is the amount of overlap
between particles during contact; $R_i$ and $R_j$ are the radius of particle $i$ and $j$; $\vec{r}_{ij}$ is



distance vector between particles; $\overrightarrow{v}_{n,ij}$ is the normal relative velocity vector. The
normal damping coefficient can be expressed as
$$d_n = \sqrt{\frac{4k_n \dfrac{m_i m_j}{m_i + m_j}(\ln \varepsilon)^2}{\pi^2 + (\ln \varepsilon)^2}}. \tag{9}$$
Where, $m_i$ and $m_j$ are the mass of particle $i$ and $j$, and $\varepsilon = 0.7$ is restitution coefficient.
The tangential force of contact is
$$\overrightarrow{F}_{t,ij} = \begin{cases} -k_t \zeta_{t,ij} \overrightarrow{\tau}_{ij} - d_t \overrightarrow{v}_{t,ij} & \left|\overrightarrow{F}_{t,ij}\right| \le \dfrac{R_i}{R_j}\left|\overrightarrow{F}_{n,ij}\right| \\[2ex] -\mu_i \left|\overrightarrow{F}_{n,ij}\right| \overrightarrow{\tau}_{ij} & \left|\overrightarrow{F}_{t,ij}\right| > \dfrac{R_i}{R_j}\left|\overrightarrow{F}_{n,ij}\right| \end{cases}. \tag{10}$$
Where, $k_t = 2 \times 10^6$ is the tangential stiffness coefficient; $\zeta_{t,ij}$ is the tangential
displacement; $\overrightarrow{v}_{t,ij}$ is the tangential relative velocity vector. The tangential damping
coefficient can be expressed as
$$d_t = 2\sqrt{\frac{m_i m_j}{m_i + m_j}k_t}. \tag{11}$$
**2.4 Splash process**

Splash processes not only serve as an indispensable part of the near-surface

particle motions, but also relate to the accuracy of emissions during particles upward
transport. There are a large number of collisions between particles and the ground.
Meanwhile, other particles will be blown up when particles hit the ground, which is
referred to as the splash process. If energy-based collision analysis is performed on a
single particle, lots of time will be consumed. Therefore, researchers parameterized
some key variables in accordance with the characteristics of splash, thereby



simplifying the problem. We assume that there are enough sand and dust particles on
the ground to splash when the particles impact the surface. If the particle collides with
the bed, we assume the rebound probability as
$$p_{reb} = 0.95(1 - e^{-\lambda v_{imp}}),$$    (12)
where $v_{imp}$ is the impact speed, and $\lambda$ is an empirical parameter in the order of 2 s/m
according to the previous study (Anderson et al., 1991). The rebound sand speed is
0.55 times of the impact sand speed, and the rebound angle $\theta_{reb}$ is 40° (Zhou et al.,
2006). Of course, at a certain speed, some new sand particles will be splashed. The
ejection number is
$$\overline{N_{ej}} = n_0 \left( 1 - \left( A - B sin\theta_{imp} \right)^2 \right) \left( \frac{v_{imp}}{\zeta \sqrt{gd_{mean}}} - 1 \right)(e^{\mu_{imp}/C} - D).$$    (13)
Where, $n_0$=0.4, $A$=0.68, $B$=0.39, $\zeta$=5, $C$=0.92, $D$=1.39 (Huang et al., 2017). $\theta_{imp}$ is the
impact angle, $\mu_{imp}$ is the ratio of impact grain size to the mean size of the bed, and
$d_{mean}$ is the mean diameter of the sand particles. The ejection angle $\theta_{ej}$ distributes
randomly between 50°~60° (Rice et al., 1995). The probability density distribution of
the initial lifting speed follows
$$p(v_{ej}) = \exp(-v_{ej} / \overline{v_{ej}}) / \overline{v_{ej}}.$$    (14)
Where, $v_{ej}$ is the ejection speed and the overbar represents a mean value (Anderson et
al., 1991; Werner, 1990). The mean ejection speed can be expressed as (Kok and
Renno, 2009)



$$\overline{v_{ej}} = \sqrt{gD_{mean}}\,\frac{\alpha_{ej}}{a}(1-\exp(-\frac{v_{imp}}{40\sqrt{gD_{mean}}})). \tag{15}$$
Moreover, the sand particles satisfy the periodic boundary condition in the direction
of streamwise and spanwise, respectively. Following the idea of Dupont et al. (2013),
aerodynamic entrainment did not consider in our model. 10000 initial particles are
randomly released in the flow field (Huang, 2020), and the release height should be
lower than 0.3 m (Shao and Raupach, 1992). The results of Dupont et al. (2013)
showed that the number of released particles does not affect the final results, but only
the speed of wind-blwon sand development.
**2.5 Parameters and equivalent method of SCBs**

According to the experience of laying SCBs in practical engineering (Chang et

al., 2000) and the theoretical results of Wang and Zheng (2002), in this paper, the
height of SCB ($S_h$) is set to 10 cm, the side length of a single SCB ($S_l$) is $100\times100$ cm,
and the side thickness of the SCB ($S_n$) is set to 10 cm. The diagram of a single SCB is
shown in Fig. 1b. Moreover, in order to study the inhibition effect of the laying length
of SCBs (represented by N) on aeolian sand erosion, we set N=5~10 m, 5~20 m, 5~30
m in the simulation cases. The diagram of the laying SCBs is shown in Fig. 1a and the
main parameters of the SCBs are listed in Table 1.

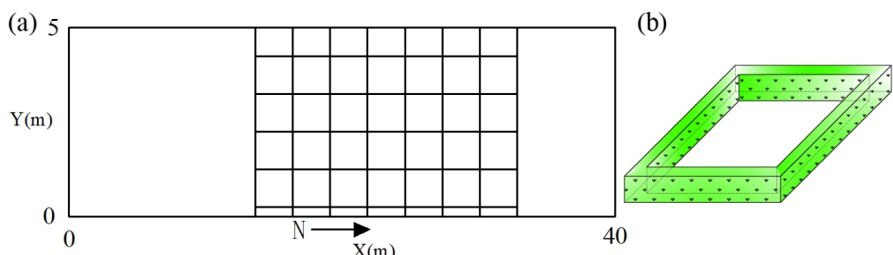






**Figure 1.** (a) The diagram of the laying SCBs. (b) The diagram of a single SCB.
The SCBs are equivalent to a volume resistance force through the resistance
coefficient and leaf area coefficient, that is, the flow in these regions will be subject to
additional resistance force, which can be expressed as
$$F_d = -C_d a |U| u_i.$$    (16)
Where, $C_d$ is the drag coefficient, $a$ is the leaf area coefficient, and $U$ is the inflow
wind speed. In the simulation, the value of $C_d$ is 0.2 according to the parameters of
Dupont et al. (2013). Nepf (2012) concluded that when the diameter of vegetation is
4-9 cm, the value of the leaf area coefficient $a$ can reach 20 m$^{-1}$. Therefore, according
to the side thickness of the SCB in this paper, the leaf area coefficient is set as 40 m$^{-1}$.
**Table 1** SCB Parameters

| Name | Symbol | Value | Unit |
|------|--------|-------|------|
| SCB height | $S_h$ | 10 | cm |
| SCB side length | $S_l$ | 100 | cm |
| SCB side thickness | $S_n$ | 10 | cm |
| laying length of SCBs | N | 5~10, 5~20, 5~30 | m |
| drag coefficient | $C_d$ | 0.2 | |
| leaf area coefficient | $a$ | 40 | m$^{-1}$ |

**2.6 Calculation parameters**



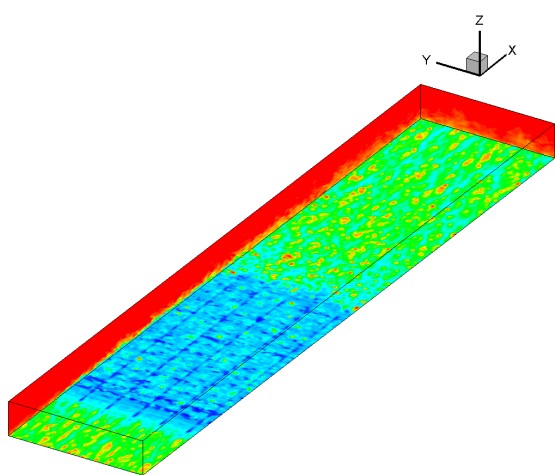


**Figure 2.** Schematic diagram of three-dimensional wind-blown sand in presence of SCBs.

261  Wind tunnel experiments conducted by Shao and Raupach (1992) indicated that

262 a complete "overshoot" had more than 10 m in streamwise (Huang et al., 2014; Ma

263 and Zheng, 2011). In Fig. 2, the computational domains are $Lx$=40 m, $Ly$=5 m, $Lz$=2

264 m in streamwise, spanwise and wall-normal directions, respectively. Field

265 experiments conducted by Baas and Sherman (2005) showed that the mean lateral size

266 of sand streamers is about 0.2 m. In order to capture this structure, the mesh spacing is

267 0.1 m and 0.1 m in streamwise and spanwise, respectively. Besides, in the near wall

268 region, the logarithmic stretching has been adopted to ensure the precision. The mean

269 and minimum mesh spacing in the vertical direction is 0.025 and 0.005 m,

270 respectively. Therefore, the grids of streamwise, spanwise and vertical directions are

271 400×100×80, respectively. The sand diameter satisfies the normal distribution: mean

272 diameter equals to 200 μm and the variance is ln(1.2). We first simulate the clean air

273 flow in presence of SCBs for 30 seconds to get fully developed. Then we add the sand





particles in the simulations to develop the sand-laden flow. After the wind-blown sand
flow becomes saturated, the simulations last another 20 seconds to do the statistics.
The fluid time step $\Delta t_s$=0.0002 s, and the particle time step $\Delta t_p$=0.00005 s. The density
of sand grain is 2650 kg/m$^3$, and the density of air 1.225kg/m$^3$. The main calculation
parameters are listed in Table 2.
**Table 2** Main Simulation Parameters

| Name | Symbol | Value | Unit |
|---|---|---|---|
| streamwise computational domain | $Lx$ | 40 | m |
| spanwise computational domain | $Ly$ | 5 | m |
| wall-normal computational domain | $Lz$ | 2 | m |
| fluid time step | $\Delta t_s$ | 0.0002 | s |
| friction wind speed | $u_*$ | 0.3, 0.44, 0.6 | m/s |
| particle time step | $\Delta t_p$ | 0.00005 | s |
| sand density | $\rho_a$ | 2650 | kg/m$^3$ |
| air density | $\rho_f$ | 1.225 | kg/m$^3$ |
| gravity | $g$ | 9.81 | m/s$^2$ |


**3. Model validations**

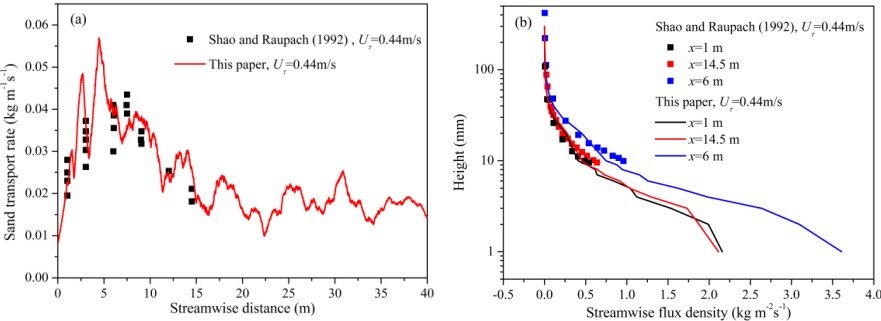

**Figure 3.** (a) The spatial variation of the streamwise sand transport rate in the sand-laden flow. (b)
The streamwise sand transport rate density with the height at the three flow direction positions.

The verification of the flow field part of the program is covered in great detail in

our previous works (Huang, 2020). In this section, we will verify the validity of the





model from the following three aspects. Sand transport rate is an important physical
quantity in the wind-blown sand, which is the embodiment of the sediment carrying
capacity of the flow field (Zheng, 2009). Therefore, without considering the SCBs, we
first compare the spatial variation of the sand transport rate in the sand-laden flow
with the experimental results of Shao and Raupach (1992). The sand transport rate is
calculated according to the formula
$$q = \sum_{z=0}^{z=H} \sum_{y=0}^{y=M} m(x) \,/\, \Delta x \,/\, \Delta t_s. \tag{17}$$

$m(x)$ represents the sand mass in the range of flow direction $x$ to $x+\Delta x$, and $\Delta x$ is the
grid size in the flow direction. And judge whether the wind-blown sand flow is
saturated by the change of the sediment transport at a certain streamwise position. The
condition for judging saturation is given by Ma and Zheng (2010). The wind tunnel
experimental results of Shao and Raupach (1992) showed that the streamwise sand
transport rate increased first, then decreased until it was stable, which is called the
"overshoot" phenomenon (Anderson and Haff, 1991; McEwan and Willetts, 1991).
Fig. 3a shows the comparison between the simulation results of the sand transport rate
along the flow direction with the experimental results of Shao and Raupach (1992)
under the same friction wind speed. As can be seen from Fig. 3a, our simulation
results also show this phenomenon. However, unlike the other numerical simulation
results (Huang et al., 2014; Ma and Zheng, 2011), our sediment transport rate results
have an obvious fluctuation characteristic that are not smooth curves, which may be
caused by the turbulence intermittency unique to the three-dimensional wind-blown



sand model. What's more, we give the distribution results of the streamwise sand
transport rate density with the height at the three flow direction positions, which are
compared with the experimental results. From Fig. 3b, we can see that the distribution
of the streamwise sand transport rate density with the height follows the trend of
exponential decline, and the sand transport rate density at $x = 6$ m is significantly
higher than that at $x = 1$ m and $x = 14.5$ m, which is consistent with the experimental
results of Shao and Raupach (1992). This is because the flow direction of $x = 6$ m is in
the peak region of the "overshoot" phenomenon, while the flow direction of $x = 1$ m
and $x = 14.5$ m is in the rising region and stable region, respectively. Due to the
massive accumulation of sand particles exist near the surface (0-20 mm), thus the
concentrations cannot be measured easily. In Fig. 3b, our simulation results can also
show that the distribution of the streamwise sand transport rate density with the height
below 10 mm still satisfies the trend of exponential decline. However, at a height of
2~3 mm, there is a slight change in this trend, that is, the rate of increase in the sand
transport rate density has slowed down, which is not revealed in the experimental
results. Due to the limitations of the large eddy simulation, the simulation results near
the wall may be distorted, so this part needs to be further verified by the experiments.





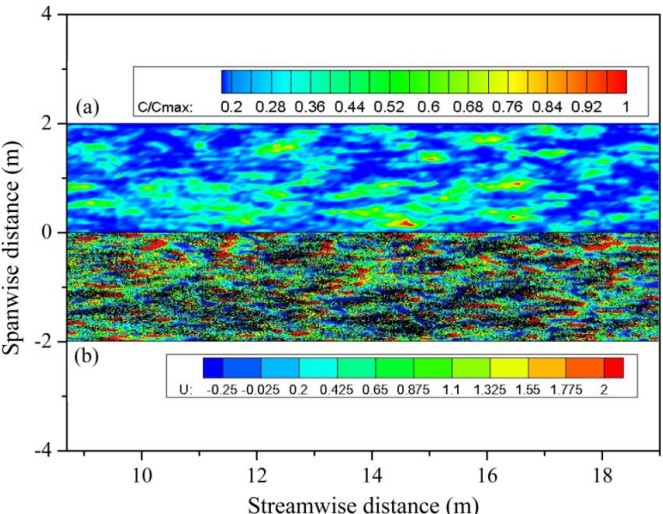


**Figure 4.** (a) The top view of the sand streamer concentrations, where C represents the particle

concentrations, $C_{max}$ represents the maximum particle concentrations. (b) The top view of the

whole particle positions and the streamwise velocity diagram of flow field with the height of 0.005

m, and the y coordinates are correspondingly shifted down by 2, where the black dots represent

the sand particles, U represents the streamwise wind speed of the sand-laden flow ($u_\tau$=0.3 m/s).

Sand streamer, as a natural phenomenon in wind-blown sand, has been widely
concerned. Therefore, without considering the SCBs, we then analyze the morphology
of sand streamer and its relationship with the flow field. In the meantime, the
air-borne particle concentration within a certain area can be calculated as
$$C = \sum_{z=0}^{z=H} \sum_{y=0}^{y=M} \sum_{x=0}^{x=L} m(x) \, / \, Lx \, / \, Ly \, / \, Lz. \tag{18}$$
Fig. 4a is the top view of particle-aggregation morphology in the stable stage of
sand-laden flow. It can be seen from Fig. 4a that the concentration of sand particles is
intermittent in both streamwise and spanwise directions. Moreover, we can see clearly



that the morphological characteristics of sand streamer are consistent with the
observations of Baas and Sherman (2005), that is, it is up to a few meters in the
streamwise direction and about 0.2 meters in the spanwise direction. Our model can
reproduce the "sand streamer" phenomenon in wind-blown sand well. Here, we need
to point out that the intermittence of turbulence complicates the particle movement,
especially when multiple streamers are connected end to end as well as the
concentration is close enough, there will existing the super sand streamers up to tens
of meters long. Whether in the sand-laden flow or the other two-phase flows,
researchers are generally concerned about the aggregation of particles. We plot the
position of particles and the streamwise velocity of flow field in Fig. 4b, and notice
that most particles are assembled in the low-speed streaks, which is consistent with
the conclusion of the other particle-laden flows (Lee and Lee, 2015; Richter, 2015).

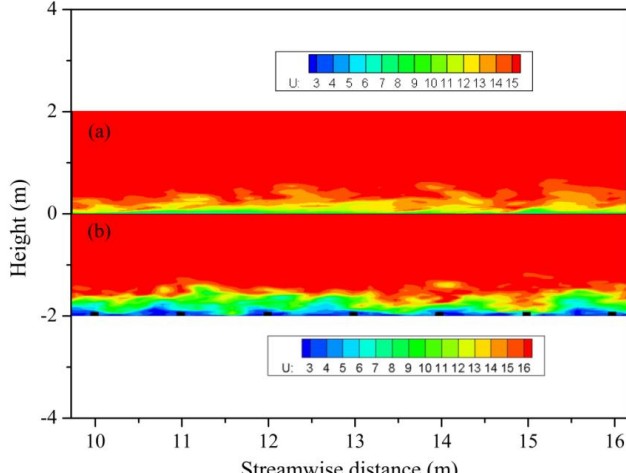


**Figure 5.** The side view of X-Z plane streamwise velocity before (a) and after (b) containing the
SCBs ($u_\tau$=0.6 m/s, N=5~20 m, $y$=0 m). The y coordinates are correspondingly shifted down by 2





in the case (b).

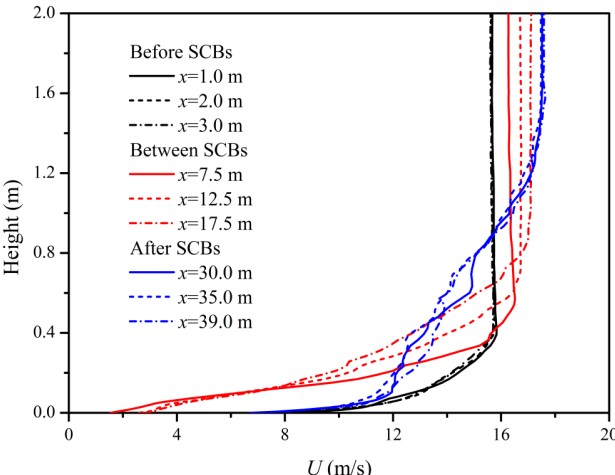


**Figure 6.** The wind speed profiles of different streamwise positions in the clean air flow
containing the SCBs ($u_\tau$=0.6 m/s, N=5~20 m).

Finally, we verify the difference of the velocity profile as well as the surface

roughness in the clean air flow with and without the SCBs. In the previous studies, the
wind speed and surface roughness near the SCBs were studied well (Dong et al., 2000;
Qu et al., 2007; Wang et al., 1999). These works all pointed out that laying the SCBs
can effectively increase the surface roughness and reduce the wind speed near the
surface, so as to play a role in inhibiting the wind-blown sand and fixing the sand
particles. Fig. 5a and 5b are the tangent plane (X-Z plane) of the streamwise wind
velocity without and with the SCBs, respectively. It can be seen intuitively that the
existence of the SCBs reduces significantly the surface wind speed, and increases the
boundary layer thickness of the flow field. In order to reveal the difference
quantitatively, we plot the wind speed profiles of different streamwise positions in the



clean air flow containing the SCBs in Fig. 6. The selected positions are $x$ = 1, 2, 3 m
in front of the SCBs, $x$ = 7.5, 12.5, 17.5 m in the area containing the SCBs, and $x$ = 30,
35, 39 m behind the SCBs. We can see that the wind speed profiles at the three
positions in front of the SCBs are basically the same. In the area containing the SCBs,
the existence of the SCBs reduces the surface wind speed and increases the thickness
of the boundary layer (equivalent to increasing the surface roughness) as well as the
incoming wind speed outside the boundary layer. Moreover, the longer the SCBs are,
the thicker the boundary layer will be, and the incoming wind speed outside the
boundary layer will also increase more. The flow field behind the SCBs may be
complicated by the influence of the attached vortex generated by the SCBs, but the
overall trend is the same and the boundary layer thickness remains consistent. These
results are qualitatively consistent with the existing conclusions (Dong et al., 2000;
Qu et al., 2007; Wang et al., 1999), which indicates that our model has effectively
introduced the SCBs module. In the following section, we will reveal more about the
influence of the laying length on the wind field and its inhibition effect on the
wind-blown sand.
**4. Results and Discussion**
**4.1 The influence of the SCBs on the clean air flow**



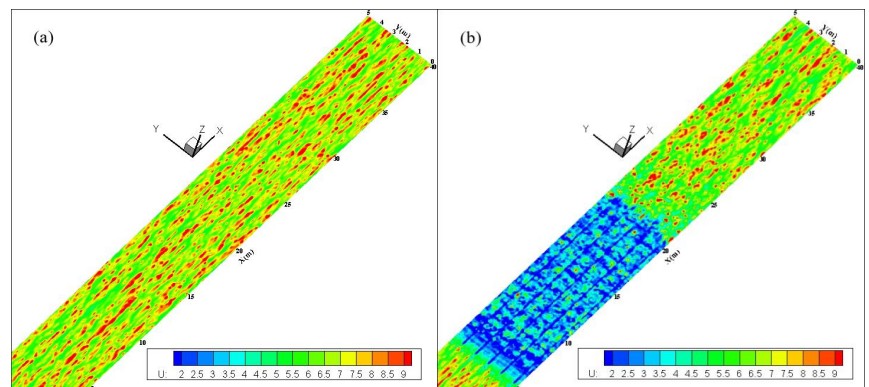

**Figure 7**. The top view of X-Y plane streamwise velocity without (a) and with (b) the SCBs

($z$=0.005 m, $u_\tau$=0.6 m/s, N=5~20 m).

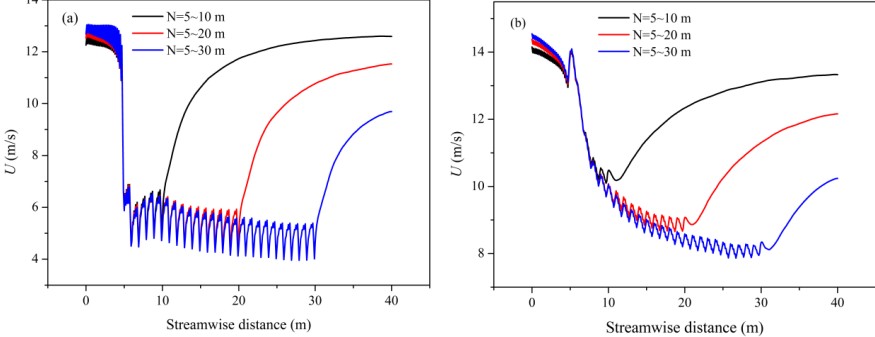

**Figure 8.** The streamwise wind speed in the clean air flow containing the SCBs at the height of

0.1 m (a) and the height of 0.2 m (b), $u_\tau$=0.6 m/s, N=5~10 m, 5~20 m, 5~30 m.

Fig. 7a and 7b show the presence of the SCBs destroys the original streaks of the

clean air flow and decreases the wind speed. The wind speed in the central area of a

single SCB is significantly higher than that in the surrounding area, showing a block

of velocity distribution characteristics. Although the wind speed behind the SCBs will

recover rapidly, there is a significant difference between the newly formed streaks and

the original streaks of the flow field, that is, the streamwise scale of the streaks behind





the SCBs is significantly shorter than before. The variation of streamwise wind speed
at the different laying length cases (N=5~10 m, 5~20 m, 5~30 m) under the same
friction wind speed ($u_\tau$=0.6 m/s) was plotted in Fig. 8, where Fig. 8a corresponds to
the wind speed at the height of 0.1 m, and Fig. 8b corresponds to the wind speed at
the height of 0.2 m. It can be seen from Fig. 8 that the wind speed in the SCBs
decreases in a process of oscillation. And behind the SCBs, the wind speed gradually
increases and returns to stability. The trend of wind speed reduction in the SCBs is
consistent with the existing experimental results (Xu et al., 1982). The difference is
that the reduction process of the wind speed around the SCBs was oscillatory
attenuation instead of continuous decrease, which is not revealed in the previous
simulation results (Bo et al., 2015). Moreover, when the incoming wind speed is
stable, the longer the laying lengths, the lower the wind speed in the stable stage
behind SCBs. This is very useful information. On this basis, we can obtain the
relationship between the laying length of the SCBs and the wind speed in the stable
stage according to an actual situation. For example, reduce the wind speed in the
stable stage to the impact threshold or the aerodynamic threshold on both sides of the
desert highway, so as to determine the minimum laying length of the SCBs and save
the laying cost. This is a potential application of our model and needs to be further
verified by the experiments.
**4.2 Effect of sand particles on the flow field and its aggregation location**





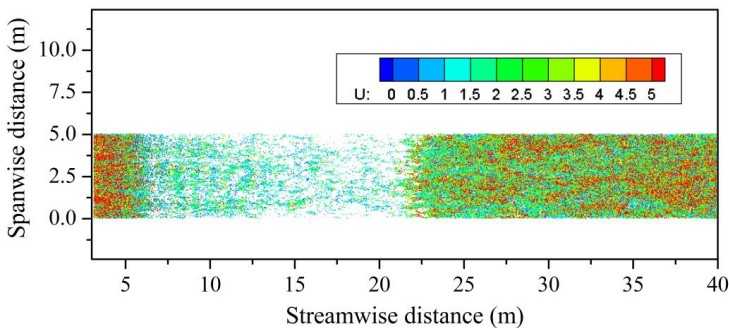


**Figure 9.** The top view of the particle positions of the wind-blown sand in presence of SCBs,

where U represents the speed of the particles ($u_\tau$=0.6 m/s, N=5~20 m).

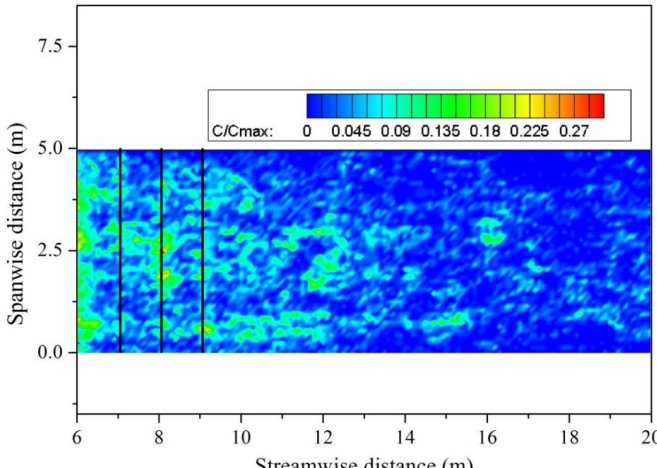


**Figure 10.** The top view of the sand concentrations in the regions of the SCBs, where C represents

the particle concentrations, $C_{max}$ represents the maximum particle concentrations ($u_\tau$=0.6 m/s,

N=5~20 m). The black lines represent the schematic diagram of the side of SCBs.

Then, sand particles were added to the clean air flow field in presence of SCBs

to fully develop and reach stability. The top view of the particle positions of the

wind-blown sand after reaching a stable state is shown in Fig. 9. From Fig. 9, we can

see that when the wind-blown sand pass through the SCBs, the particle number



obviously decreases gradually, and the inhibition effect of the SCBs on the
wind-blown sand can be visualized. Moreover, the motion of sand particles behind the
SCBs returns to a complete wind-blown sand movement. We then plot the sand
concentrations of the region in presence of SCBs in Fig. 10. Combining the laying
position of the SCBs as well as the corresponding sand concentrations, we can clearly
see that the concentration of sand particles near the side of SCBs is higher than that in
its central region, which is consistent with the conclusion of Xu et al. (2018). On the
one hand, the wind speed near the side of SCBs is low, and the drag force of the sand
particles in these areas will be significantly reduced, so that the sand particles will
accumulate or deposit in these regions. On the other hand, the wind speed in the
central area of every single SCB is significantly higher than that in the surrounding
area, so that the sand particles are not easy to gather or fall in these regions. This
explains why the side of the SCBs tends to be buried in the sandy land and loses its
effect after long working hours.

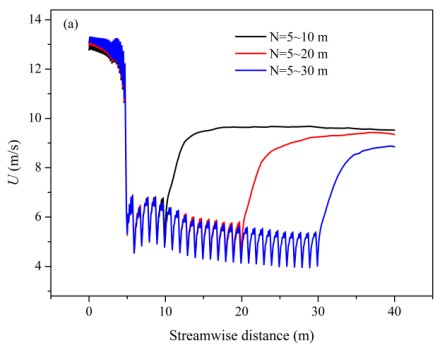 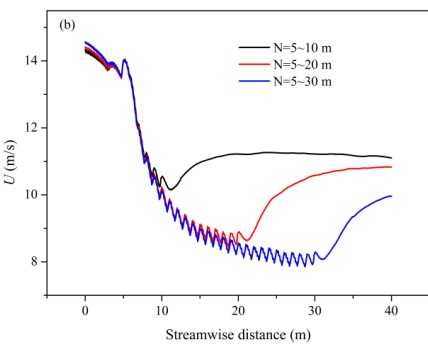




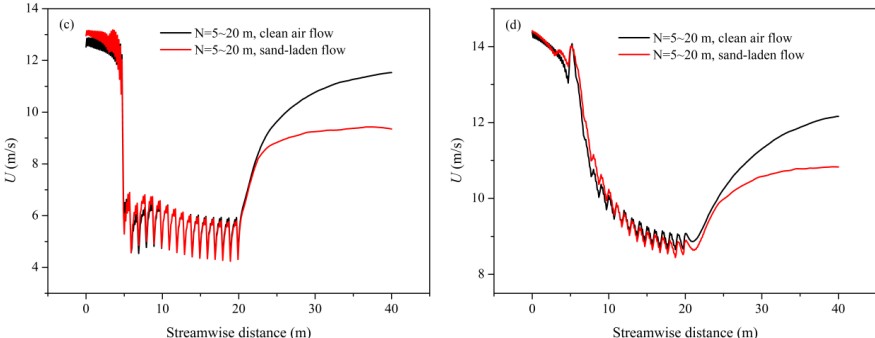


**Figure 11.** The streamwise wind speed in the sand-laden flow containing the SCBs at the height of

0.1 m (a) and the height of 0.2 m (b), $u_\tau$=0.6 m/s, N=5~10 m, 5~20 m, 5~30 m. The comparison of

the streamwise wind speed between the clean air flow and the sand-laden flow at the height of 0.1

m (c) and the height of 0.2 m (d), $u_\tau$=0.6 m/s, N=5~20 m.

Furthermore, we analyze the effect of sand particles on the wind speed in the
sand-laden flow. The streamwise wind speed of the sand-laden flow at the different
laying length cases (N=5~10 m, 5~20 m, 5~30 m) under the same friction wind speed
was plotted in Fig. 11a and 11b. Meanwhile, for the convenience of comparison, the
streamwise wind speed under the same laying length (N=5~20 m) in the sand-laden
flow and the clean air flow were plotted in Fig. 11c and 11d. Fig. 11a and 11c
correspond to the wind speed at a height of 0.1 m, while Fig. 11b and 11d correspond
to the wind speed at a height of 0.2 m. From Fig. 11a-d, we can see that the wind
speed in the SCBs of the sand-laden flow still decreases in a process of oscillation.
The streamwise wind speed behind the SCBs in the sand-laden flow is significantly
lower than that in the clean air flow. Obviously, the presence of sand particles indeed
reduces the wind speed. However, the change of wind speed in the SCBs between the




sand-laden flow and the clean air flow is not obvious, because there are fewer sand
particles in the SCBs than behind the SCBs, which has less effect on the overall wind
speed.
**4.3 Effect of laying length on the sand transport rate**

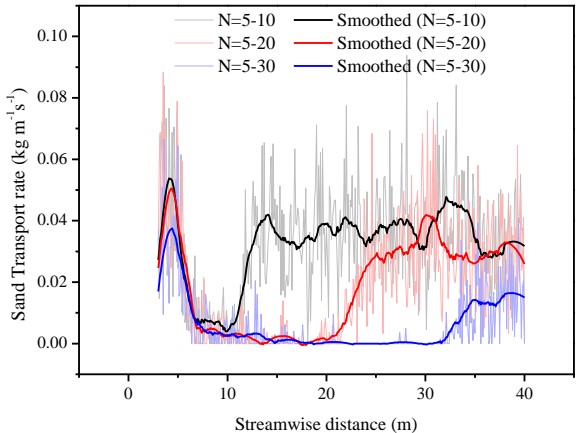


**Figure 12.** The streamwise sand transport rate in the different laying length cases ($u_\tau$=0.6 m/s,
N=5~10 m, 5~20 m, 5~30 m). Dark lines are the result of smoothing.
Here, the effect of the different laying length cases (N=5~10 m, 5~20 m, 5~30 m)
on the sand transport rate under the same friction wind speed was plotted in Fig. 12. It
can be seen from Fig. 12 that the sand transport rate in the SCBs is very low, and with
the increase of the laying length, the sand transport rate in the SCBs will be lower and
lower. In the case of N=5~30 m, we can even see that the sand transport rate in some
regions has been reduced to zero. Therefore, this result once again shows that the
laying length of the SCBs can be optimized, and we can reduce the laying cost while
keep the effect of the SCBs unchanged. Especially on both sides of the desert highway,

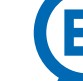

our model can give the minimum laying length according to the actual parameters. At
the same time, we notice that the sand transport rate will increase rapidly and then
reach to stable state behind the SCBs. And it is obvious that when N=5~30 m, the
value of sand transport rate at the stable stage behind the SCBs is significantly lower
than the other results of N=5~10 m and N=5~20 m. We also notice that the longer the
laying lengths, the lower the sand transport rate in the stable stage behind the SCBs.
This result is corresponding to the result of Fig. 8. Our results indicate that when the
sandy land is wide, the discontinuous laying method can be considered. That is,
determine the minimum laying length first, and then determine the distance between
each minimum laying length as required. In this way, the sand transport rate can be
reduced in sections. This is another potential application of our model.
**4.4 Particle positions under different friction wind speeds**

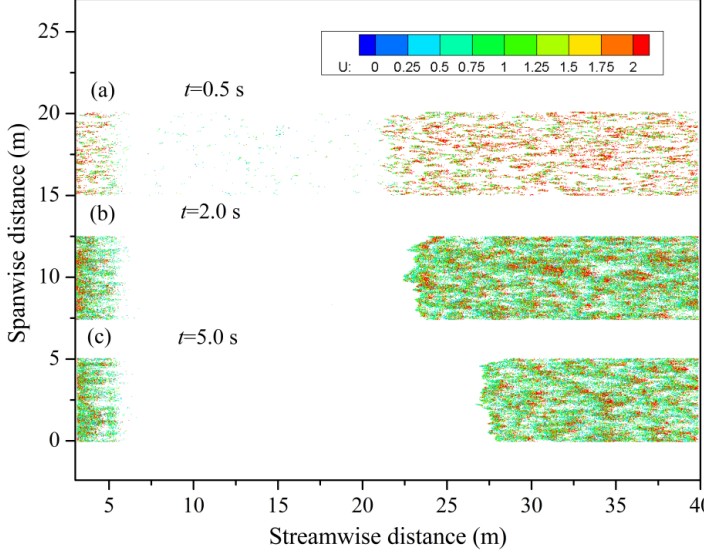


**Figure 13.** The top view of the particle positions of the wind-blown sand in presence of SCBs at





the time $t$=0.5 s (a), $t$=2.0 s (b), $t$=5.0 s (c), where U represents the speed of the particles ($u_\tau$=0.3
m/s, N=5~20 m). The y coordinates are correspondingly shifted up by 7.5 per case.

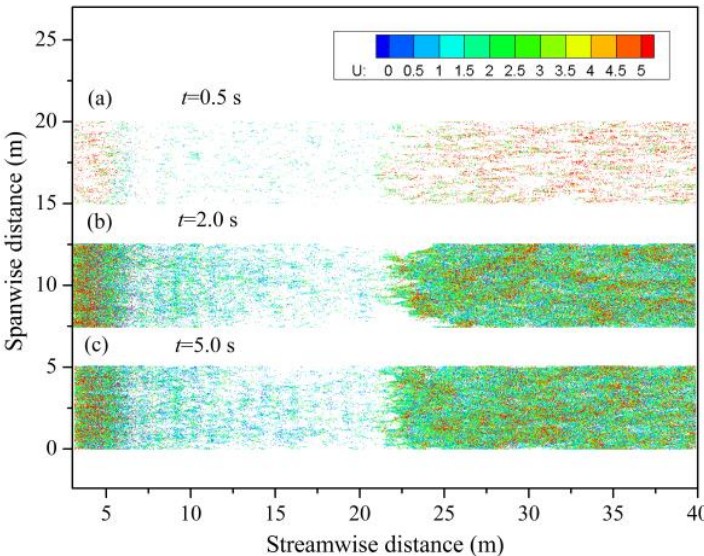


**Figure 14.** The top view of the particle positions of the wind-blown sand in presence of SCBs at
the time $t$=0.5 s (a), $t$=2.0 s (b), $t$=5.0 s (c), where U represents the speed of the particles ($u_\tau$=0.6
m/s, N=5~20 m). The y coordinates are correspondingly shifted up by 7.5 per case.

The above analysis is based on the calculation case when the friction wind speed

is 0.6 m/s, and the sand particles can easily penetrate the SCBs when the wind speed
is large. When the friction wind speed is small, the inhibition effect of the SCBs on
wind-blown sand will become more obvious, and the movement behavior of sand
particles will become different. We plot the top view of the particle positions at
different moments ($t$=0.5 s, 2 s, 5 s) when the friction wind speed is 0.3m/s and
0.6m/s in Fig. 13 and Fig. 14, respectively. The time $t$ in Fig. 13 and Fig. 14 counts
from the moment sand particles are added. The results show that when the wind speed



is small, the sand particles cannot penetrate the SCBs. There is no obvious sand
movement in the SCBs, and the stable wind-blown sand cannot be formed behind the
SCBs. With the passage of time, the wind-blown sand behind the SCBs will gradually
disappear. It is worth pointing out that aerodynamic entrainment is not considered in
our model. This is a strong limitation of our model to simulate wind erosion in
presence of SCBs. Therefore, a more reasonable situation is that when the wind speed
behind the SCBs returns to the fluid threshold, this part of the wind-blown sand
should still develop. When the wind speed is relatively small, on the one hand, the
sand particles cannot completely penetrate the regions of the SCBs, and then cannot
continuously provide the impact particles to form the wind-blown sand behind the
SCBs. On the other hand, the SCBs affect the surface wind speed behind it, thus also
affecting the continuous formation of the wind-blown sand. When the wind speed is
relatively large, the sand particles can penetrate the SCBs. And with the increase of
the laying length, although the inhibition effect on wind-blown sand is more obvious,
the stable wind-blown sand will still be formed behind the SCBs. We think that when
the laying length of the SCBs is fixed, whether the wind speed will decrease below
the impact threshold or the fluid threshold is the key to determine whether the sand
particles can penetrate the SCBs and form stable wind-blown sand behind the SCBs.
In order to present this phenomenon more clearly, we have animated this process, as
shown in the supplementary materials (Video 1 and Video 2). In the actual
anti-desertification projects, the minimum laying length of the SCBs can be
determined by our model according to the local maximum friction wind speed, which





is very meaningful.

## 5. Conclusions and outlook

In this paper, a three-dimensional wind-blown sand coupling model in presence
of SCBs was established. The model was verified from the following aspects: (1)
spatial distribution of sand transport rate; (2) morphological characteristics of sand
streamer from the instantaneous fields; (3) changes in the thickness of the boundary
layer before and after the SCBs. From this model, the inhibition effect of SCBs on
wind-blown sand was studied qualitatively, and the sensitivity of aeolian sand erosion
to the laying length was investigated. The results showed that the wind speed in the
SCBs of the clean air flow or the sand-laden flow both decreases in a process of
oscillation, which has not been revealed by the previous researches. Moreover, the
longer the laying lengths of the SCBs, the lower the wind speed in the stable stage
behind SCBs, and the lower the sand transport rate, which may provide the theoretical
support for the minimum laying length of SCBs in anti-desertification projects. More
importantly, we found that the concentration of sand particles near the side of SCBs is
higher than that in its central region, which is consistent with the previous research.
This explains why the boundary of the SCBs tends to be buried in the sandy land and
loses its effect after long working hours. Our results also indicated that whether the
wind speed will decrease below the impact threshold or the fluid threshold is the key
factor affecting whether sand particles can penetrate the SCBs and form stable
wind-blown sand behind the SCBs under the same conditions. Although our model
has been able to reveal the inhibition effect of the SCBs on wind-blown sand, there





are still some aspects to be improved in the future, such as the aerodynamic
entrainment, particle deposition on the SCB, and the collision between the sand
particles and the SCBs. And the size of the SCB used in our model is fixed. In the
future work, we plan to analyze the effect of different height and width of the SCB on
the aeolian sand erosion and discuss the reasons for the difference in heights between
SCB and other obstacles, such as sand fence. Another aspect worth noting is that some
additional factors such as terrain, surface roughness will affect the effect of the SCBs
in the anti-desertification project, so the influence of these factors should be
considered in the future. The significance of our work is to analyze some results
which seemingly simple but lack of theoretical basis from the perspective of
turbulence through this model.

**Acknowledgments**
This research was supported by grants from the National Natural Science Foundation
of China (Grant number 12002119), Opening Foundation of MOE Engineering
Research Center of Desertification and Blown-sand Control, Beijing Normal
University (2021-B-4). The author expresses sincere appreciation to the supports. The
author would like to thank the Center for Analysis and Prediction of Storms (CAPS)
at the University of Oklahoma for providing the original ARPS code.

**Code and Data availability.** All relevant code and data used to generate the



figures in this paper can be accessed using the following email: hjhuang@usst.edu.cn.

**Competing interests.** The authors declare that there are no competing interests

**Author contributions.** HJ performed the programming, analyzed the results, and
wrote the paper.

**Video supplement.** Video 1 and Video 2 can be downloaded at the following link:
https://doi.org/10.5281/zenodo.6937805

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
