# Peer review of "Modeling the Inhibition Effect of Straw Checkerboard"

_EGUsphere, 2022_

## Referee Comment (RC1)

**Review of the paper** :
Modeling the inhibition effect of straw checkerboard barriers on wind-blown sand
by Haojie Huang

The author presents a numerical study of the impact of straw checkerboard barriers (SCB) on the aeolian sand transport. Large eddy simulation are performed in presence of SCB with different laying lengths. Saltation is enhanced through the splash process. The inhibition effects of SCB on the sand transport is investigated. When the layer length increases, the wind speed and the sand transport rate decreases. The study help to understand the impact of SCB and may be useful for anti desertification projects.

This paper bring a few new insights on the effects of SCB. I then recommend to accept this article with major revisions.

The article should be proofread to correct English. Some sentences are not correct. For example in the sentence line 21-23, there is no verb. I do not understand the sentence 95-96.

 Basic conventions such as: do not put a title at the end of a page should be respected. A figure legend should be on the same page as the corresponding figure. Put a space between the paragraph number and the title (line 183).

Section 2.1:
- line 137: the force Fi (equation 1) should be detailed. The formula and a reference should be given.
- line 152: on the ground, the author say that a rigid condition is used. Is the velocity put to zero or are the ARPS wall function used? How is the SBC taken into account? It is not possible to construct a boundary with sharp angles with the code ARPS.
-line 156: the variable D is not defined.
- line 161: just give the reference not a figure number of another paper to avoid confusion with the figures of the present paper.

Section 2.2:
- line 177: the particle Reynolds number is defined and do not appear in the equations.

Section 2.4:
- line 236: wind-blwon

Section 2.5:
- The figure 1 is not clear. The variables N, SCB side length, SBC side thickness and the laying length of SCB should appear on the figure. The side length is defined as 100x100cm in the text (line 240) and as 100cm in the table.

Section 2.6:
- Figure 2 is not clear. Which quantity is presented? There is no legend. The size of the mesh and the checkerboards could be plotted instead.
- The first sentence of the paragraph (line 261-263) does not seem to belong to this section, but to the section 3.
- The grid step should be added in the Table2.

Section 3:
- This section should be divided into two subsection: Particle field validation and Velocity field validation.
- Line 314: the work 'direction' seems not correct. Do you mean the location of the section?
- Line 317: suppressed the word 'exist'.
- Line 335: Define H,M,L. Are M and L equal to the grid step? Is H equal to Lz?
- Why is the friction velocity equal to 0.3 m/s in the figure 4 and to 0.6 m/s in the figure 5?
- The author could complete the analysis by plotting the recirculation zones that should appear inside the SBS.
- The mesh is very stretched near the wall with a ratio 5/100 between dz and dx. This may create diffusion problem. The authors should present mean and rms velocity profiles of the boundary layer without the SBS to validate the velocity field.

Section 4.1:
- Section 4.1 discuss of the SBSs influence on the flow. It was already the subject of the precedent paragraph and of the figures 5 and 6. These results should be put into the same paragraph.
- Line 395: the velocity seems to be smaller and not higher.

Section 4.2:
- I don't understand what is presented on the figure 9. The author speaks about 'particle position' but I don't see any particles.
- Figure 10: Is the concentration obtained at the a given height or is it the total concentration at all the height of the SBS?

Section 4.4:
- The authors presents instationary results and provide no comments about the time evolution.
- The initial state is not a realistic. Particles are not induced by the incipient motion but randomly dispatched in the field. The time evolution is then not really meaningful and so the author should only present stationary results.

---

## Author Comment (AC1)

Authors' Responses to the Comments on the Manuscript

**"Modeling the Inhibition Effect of Straw Checkerboard Barriers on Wind-blown Sand"**

**General Response to the Comments and the Suggestions:**

According to the suggestion of the **Referee1**' comments, we give a substantial revision of the original manuscript such that a clear description of the research is displayed in the revised version. I hope that our efforts will make our works recognized. The authors also would like to kindly thank the **Referee1** for give us a chance to modify this article. The detailed responses are as follows.

**Responses to Comments of Reviewer#1:**

**Main comments:**

*The author presents a numerical study of the impact of straw checkerboard barriers (SCB) on the aeolian sand transport. Large eddy simulation are performed in presence of SCB with different laying lengths. Saltation is enhanced through the splash process. The inhibition effects of SCB on the sand transport is investigated. When the layer length increases, the wind speed and the sand transport rate decreases. The study help to understand the impact of SCB and may be useful for antidesertification projects.*

*This paper bring a few new insights on the effects of SCB. I then recommend to accept this article with major revisions.*

*The article should be proofread to correct English. Some sentences are not correct. For example in the sentence line 21-23, there is no verb. I do not understand the sentence 95-96.*

*Basic conventions such as: do not put a title at the end of a page should be respected. A figure legend should be on the same page as the corresponding figure. Put a space between the paragraph number and the title (line 183).*

**Authors' main Response**

Thanks for Referee1's carefully and objectively reviewing of the manuscript, and the comment: '**useful for anti-desertification**', '**a few new insights**', this is a great affirmation to the authors' work. Referee has made a series of instructive suggestions which help us to improve the integrity of the introduction and enhance the readability and the profundity of the paper.

The authors have modified the original manuscript carefully on the grammar and writing style, and we have made many changes in the revised version, such as, we

have modified the sentence "Moreover, the longer the laying lengths, the lower the wind speed in the stable stage behind SCBs, and the lower the sand transport rate." as "Moreover, the longer the laying lengths are, the lower the wind speed and the sand transport rate in the stable stage behind SCBs will be.", we have modified the sentence "Moreover, since the actual three-dimensional SCB is simplified into two-dimensional plane with only streamwise direction and vertical direction. And the impact of this simplification is uncertain." as "Moreover, since the three-dimensional SCB is simplified into a two-dimensional plane with only the streamwise direction and vertical direction, the impact of this simplification is uncertain.", please see lines 21-22, 94-97, ...... for detailed information. English is always a tough challenge for us who are non-native English speakers. We also had the article retouched using some proof-reading services. Perhaps it can improve our writing ability as soon as possible.

We also have revised the manuscript according to the notes "do not put a title at the end of a page should be respected" and "A figure legend should be on the same page as the corresponding figure" mentioned by the reviewer.

**Item 1:**

***Section 2.1:***
***(1) Line 137:*** *the force Fi (equation 1) should be detailed. The formula and a reference should be given.*
***(2) Line 152:*** *on the ground, the author say that a rigid condition is used. Is the velocity put to zero or are the ARPS wall function used? How is the SBC taken into account? It is not possible to construct a boundary with sharp angles with the code ARPS.*
***(3) Line 156:*** *the variable D is not defined.*
***(4) Line 161:*** *just give the reference not a figure number of another paper to avoid confusion with the figures of the present paper.*

**Authors' Response 1:**

Thank you for your suggestion and question.
**(1):**
The authors have added the detailed description about $F_i$ in the revised manuscript.

$F_i = F_{pi} + F_{di}$ is the main feedback force, including the feedback force provided by sand particles ($F_{pi}$, as shown in Section 2.2) and the SCBs ($F_{di}$, as shown in Section 2.5).

In Section 2.2, we introduced the feedback force provided by sand particles:

$$m_p \frac{d^2 x}{dt^2} = F_{px} = \frac{C_D \pi D^2 \rho_f}{8} (\tilde{u} - \frac{dx}{dt})^2 + F_{nx} + F_{sx},$$

$$m_p \frac{d^2 y}{dt^2} = F_{py} = \frac{C_D \pi D^2 \rho_f}{8} (\tilde{v} - \frac{dy}{dt})^2 + F_{ny} + F_{sy},$$

$$m_p \frac{d^2z}{dt^2} = F_{pz} = -\frac{\pi g \rho_p D^3}{6} + \frac{C_D \pi D^2 \rho_f}{8}(\tilde{w} - \frac{dz}{dt})^2 + F_{nz} + F_{sz}.$$

In Section 2.5, we introduced the feedback force provided by SCBs:

$$F_{di} = -C_d a |U| u_i.$$

And we have explained "$i$ = 1, 2 and 3 correspond to the streamwise, spanwise and wall-normal directions" in the line 137.

**(2):**

SCBs are not added to ARPS in the form of a boundary. The resistance force source method has been used to equate the effect of SCBs.

In Section 2.5, we introduced this method:

"The SCBs are equivalent to a volume resistance force through the resistance coefficient and leaf area coefficient, that is, the flow in these regions will be subject to additional resistance force, which can be expressed as

$$F_{di} = -C_d a_d |U| u_i,$$

where, $C_d$ is the drag coefficient, $a_d$ is the leaf area coefficient, and $U$ is the inflow wind speed."

As far as the authors know, ARPS does not provide wall functions, nor do we add them here. Large eddy simulation without wall correction is also acceptable, such as our previous work (*Huang, H. J.: Modeling the effect of saltation on surface layer turbulence, Earth Surf. Proc. Land. 45(15), 3943-3954, 2020.*) and other research group works (*Dupont S, Bergametti G, Marticorena B, Simoëns S.: Modeling saltation intermittency. J. Geophys. Res. Atmos. 118(13), 7109-7128, 2013.*).

**(3):**

Thank you for your reminder.

We have modified the sentence "$z_0=D/30$ is the aerodynamic surface roughness" as "$z_0=d_{mean}/30$ is the aerodynamic surface roughness, and $d_{mean}$ is the mean diameter of the sand particles".

**(4):**

Thank you for your reminder.

The authors have corrected it. Please see lines 160-163 in the revised manuscript.

", the LWS method (Lund et al., 1998) is applied to the inlet condition and the recycling plane at $x_{ref}/Lx$=12.5% (Inoue and Pullin, 2011). $x_{ref}$ =5 m is the position of the recycling plane, and $Lx$=40 m is the total length of the flow direction."

**Item 2:**

*Section 2.2:*
*(1) **Line 177:** the particle Reynolds number is defined and do not appear in the equations.*

**Authors' Response 2:**

$C_D = (0.63 + 4.8 / \mathrm{Re}_p^{0.5})^2$ is the drag coefficient of sand particle, but the display is incomplete in the original manuscript. We have supplemented it. Thank you for your reminder.

**Item 3:**

***Section 2.4:***
***(1) Line 236:*** *wind-blwon*

**Authors' Response 3:**

The author has corrected it.

**Item 4:**

***Section 2.5:***
*(1) The figure 1 is not clear. The variables N, SCB side length, SBC side thickness and the laying length of SCB should appear on the figure. The side length is defined as 100x100cm in the text (line 240) and as 100cm in the table.*

**Authors' Response 4:**

Thank you for your suggestion. We have made the following modifications to Figure 1:

[Figure]

**Figure 1.** (a) The diagram of the laying SCBs. (b) The diagram of a single SCB.
We also have corrected the definition of SCBs side length. Please see line ?? in the revised manuscript.
", the side length of a single SCB ($S_l$) is set to 100 cm, and the side thickness of the SCB ($S_n$) is set to 10 cm."

**Item 5:**

***Section 2.6:***
*(1) Figure 2 is not clear. Which quantity is presented? There is no legend. The size of*

*the mesh and the checkerboards could be plotted instead.*
*(2) The first sentence of the paragraph (line 261-263) does not seem to belong to this section, but to the section 3.*
*(3) The grid step should be added in the Table2.*

**Authors' Response 5:**

**(1):**

In Figure 2, we have added a legend and the coordinates in three directions. As the flow velocity at the SCBs position is relatively low, it can also be well distinguished in the figure.

[Figure]

**Figure 2.** Schematic diagram of three-dimensional wind-blown sand in presence of SCBs.

**(2):**

"Wind tunnel experiments conducted by Shao and Raupach (1992) indicated that a complete "overshoot" had more than 10 m in streamwise (Huang et al., 2014; Ma and Zheng, 2011)." What we want to express in this sentence is that our flow direction computing domain is long enough.

**(3):**

In the original manuscript, we mentioned the grid information:

"To capture this structure, the mesh spacing is $N_x$=0.1 m and $N_y$=0.05 m in the streamwise and spanwise directions, respectively. In addition, in the near-wall region, logarithmic stretching has been adopted to ensure precision. The mean and minimum mesh spacing in the vertical direction is $N_z$=0.025 m and $N_{zmin}$=0.005 m, respectively. Therefore, the grids of the streamwise, spanwise and vertical directions are 400×100×80, respectively.".

Following the suggestion of the reviewer, we have added the grid information in the Table 2.

**Table 2** Main Simulation Parameters

| Name | Symbol | Value | Unit |
|---|---|---|---|
| streamwise computational domain | $Lx$ | 40 | m |
| spanwise computational domain | $Ly$ | 5 | m |
| wall-normal computational domain | $Lz$ | 2 | m |
| fluid time step | $\Delta t_s$ | 0.0002 | s |
| friction wind speed | $u_*$ | 0.3, 0.44, 0.6 | m/s |
| particle time step | $\Delta t_p$ | 0.00005 | s |
| sand density | $\rho_a$ | 2650 | kg/m$^3$ |
| air density | $\rho_f$ | 1.225 | kg/m$^3$ |
| gravity | $g$ | 9.81 | m/s$^2$ |
| streamwise mesh spacing | $N_x$ | 0.1 | m |
| spanwise mesh spacing | $N_y$ | 0.05 | m |
| wall-normal mean mesh spacing | $N_z$ | 0.025 | m |

**Item 6:**

*Section 3:*
*(1) This section should be divided into two subsection: Particle field validation and Velocity field validation.*
*(2) Line 314: the work 'direction' seems not correct. Do you mean the location of the section?*
*(3) Line 317: suppressed the word 'exist'.*
*(4) Line 335: Define H,M,L. Are M and L equal to the grid step? Is H equal to Lz?*
*(5) Why is the friction velocity equal to 0.3 m/s in the figure 4 and to 0.6 m/s in the figure 5?*
*(6) The author could complete the analysis by plotting the recirculation zones that should appear inside the SBS.*
*(7) The mesh is very stretched near the wall with a ratio 5/100 between dz and dx. This may create diffusion problem. The authors should present mean and rms velocity profiles of the boundary layer without the SBS to validate the velocity field.*

**Authors' Response 6:**

**(1), (7):**
We fully agree with you. Thank you for your suggestion.

We have divided section 3 into two parts: (3.1) particle field validation and (3.2) velocity field validation.

Section 3.2 is the model verification part of this paper, which should have contained the verification of the clean air flow. However, considering our previous work, we mentioned "The verification of the flow field part of the program is covered in great detail in our previous works (*Huang, H. J.: Modeling the effect of saltation on*

*surface layer turbulence, Earth Surf. Proc. Land. 45(15), 3943-3954, 2020).*"

The clean air flow field procedure in this article is based on the code of our previous work, and the meshing and flow field conditions are similar.

You can find the information in our previous work:

*Velocity field validation:*

[Figure]

**FIGURE 2.** The mean velocity profile (a) and the streamwise turbulence intensity profile (b), with comparisons between the simulation results from this article with the experiment results of Hutchins et al. (2009) under the same friction Reynolds number.

*Mesh information:*

*"To capture this structure, the mesh spacing is 0.1m and 0.05m in the streamwise and spanwise directions, respectively. Additionally, in the near-wall region, the logarithmic stretching has been adopted to ensure the precision. The mean and minimum mesh spacings in the vertical directions are 0.025 and 0.005m, respectively."*

So we think that this part of the verification of clean air flow field can refer to our previous work, and the focus of section 3.2 is the flow field verification after considering the SCBs.

*The mesh information* above also answers your concern about the mesh in question 7. In our previous work, the mesh stretching ratio between streamwise and vertical directions was 1:20. And, as you can see from other literature (Dupont et al., 2013), similar mesh stretching ratios for ARPS can be as high as 1:10, so the stretching ratios in this article are acceptable and do not cause divergence problems.

**(2):**

The authors have modified the sentence "This is because the flow direction of $x$ = 6 m is in the peak region of the "overshoot" phenomenon, while the flow direction of $x$ = 1 m and $x$ = 14.5 m is in the rising region and stable region, respectively." as "This is because the streamwise position of $x$ = 6 m is in the peak region of the "overshoot" phenomenon, while the streamwise positions of $x$ = 1 m and $x$ = 14.5 m are in the rising region and stable region, respectively.". Please see lines 316-318 in the revised manuscript.

**(3):**

We have removed the word "exist".

**(4):**

We have modified the Eqs. (17) and (18).

The original:

$$q = \sum_{z=0}^{z=H} \sum_{y=0}^{y=M} m(x) \, / \, \Delta x \, / \, \Delta t_s.$$

$$C = \sum_{z=0}^{z=H} \sum_{y=0}^{y=M} \sum_{x=0}^{x=L} m(x) \, / \, Lx \, / \, Ly \, / \, Lz.$$

The revised:

$$q(x) = \sum_{z=0}^{z=Lx} \sum_{y=0}^{y=Ly} m(x, y, z) \, / \, \Delta x \, / \, \Delta t_s.$$

$$C = \sum_{z=0}^{z=Lz} \sum_{y=0}^{y=Ly} \sum_{x=0}^{x=Lx} m(x, y, z) \, / \, Lx \, / \, Ly \, / \, Lz.$$

**(5):**

In this paper, three different friction wind speeds (0.3, 0.44, 0.6 m/s) are set in the simulation cases. We want to show the results of different friction wind speeds as much as possible.

**(6):**

We are not sure whether you mean the recirculation zones are similar to the results in Xu et al. (2018, JGR), as shown below.

[Figure]

**Figure 4.** Contour maps of velocity in the center section. (a) Mean horizontal velocity;

If so, I'd like to clarify. The SCBs equivalence method we used is different from the method used in Xu et al. (2018, JGR), which I mentioned in the introduction: "However, the SCBs are completely equivalent to the solid as the bottom boundary condition in their model. As a nonsolid material, SCBs can be penetrated by wind in practice. It only weakens the wind speed and is thus not equivalent to a solid."

The results in Xu et al. (2018, JGR) are similar to the recirculation zones that appear in the backward-facing step flow. In this paper, the SCBs are equivalent to a volume resistance force.

Therefore, this backflow vortex phenomenon does not appear in our model, because the wind is able to pass through the SCBs (only weakened, not very strong backflow), which is also closer to reality. In addition, the side length of our SCBs is 1 m, which is twice their side length (0.5m), so this regular backflow vortex phenomenon is not significant in our model.

[Figure]

Figure. The side view of X-Z plane streamwise velocity after containing the SCBs ($u_\tau$=0.6 m/s, N=5~20 m, $y$=0 m).

**Item 7:**

*Section 4.1:*
*(1) Section 4.1 discuss of the SBSs influence on the flow. It was already the subject of the precedent paragraph and of the figures 5 and 6. These results should be put into the same paragraph.*
*(2) Line 395: the velocity seems to be smaller and not higher.*

**Authors' Response 7:**

**(1):**
Figures 5-7 all show the effect of the SCBs on the clean air flow field. Figures 5-6 focus on the increase of boundary layer thickness. Similar conclusions can be compared, so we put this part into the model validation section. The focus of Figure 7 is that the SCBs destroy the near surface turbulent structure, which is a new result and has not been revealed in the existing literature. So we put this part into the results and discussion section. I hope our explanations will satisfy you.

**(2):**
I apologize for the lack of clarity in this part of the presentation. According to the comments of the reviewer, we have zoomed in on the speed diagram inside the SCBs, and you can see:

The streamwise speed in most SCBs is higher in the central area than in the surrounding area (red box), and the streamwise speed in a few SCBs is higher in the surrounding area than in the central area (yellow box).

We have modified the sentence "The wind speed in the central area of a single SCB is significantly higher than that in the surrounding area, showing a block of velocity distribution characteristics." as "In most cases, the wind speed in the central area of a single SCB is significantly higher than that in the surrounding area, showing a block of velocity distribution characteristics."

This conclusion is also quite consistent with the actual situation. The wind speed

in the center of a single SCB is high, and then the sand particles are deposited less; the wind speed around the surrounding area is small and the sand particles are deposited more.

[Figure]

[Figure]

**Item 8:**

*Section 4.2:*
*(1) I don't understand what is presented on the figure 9. The author speaks about 'particle position' but I don't see any particles.*
*(2) Figure 10: Is the concentration obtained at a given height or is it the total concentration at all the height of the SBS?*

**Authors' Response 8:**

Thank for your question.

**(1):**

In Figure 9, these colored dots represent the sand particles. Due to the huge number of particles, it is not very clear when shown in the figure. In combination with the reviewer's question, we have selected a small part of the area for zooming, that is, the area corresponding to the red box in Figure 9. You can see the position of each particle in this area after zooming. The color of the particles in the figure represents the streamwise speed of the particles.

We updated the Figure 9 in the revised manuscript, and please see line 440 for detailed information.

[Figure]

**Figure 9.** The top view of the particle positions of the wind-blown sand in presence of SCBs, where U represents the speed of the particles ($u_\tau$=0.6 m/s, N=5~20 m).

**(2):**

According to the concentration formula, the concentration here is for all heights and does not refer specifically to a certain height.

$$C = \sum_{z=0}^{z=Lz} \sum_{y=0}^{y=Ly} \sum_{x=0}^{x=Lx} m(x, y, z) \, / \, Lx \, / \, Ly \, / \, Lz.$$

**Item 9:**

*Section 4.4:*

*(1) The authors presents instationary results and provide no comments about the time evolution.*

*(2) The initial state is not a realistic. Particles are not induced by the incipient motion but randomly dispatched in the field. The time evolution is then not really meaningful and so the author should only present stationary results.*

**Authors' Response 9:**

**(1), (2):**

I agree with you. The difference between the random initial distribution and the initial distribution based on aerodynamic entrainment is that the lift-off particles starting at different locations. This will indeed affect the initial stage of the development of wind-blown sand. When the sand flux reaches saturation, this effect is very limited. We know that the main factor that can maintain the wind-blown sand is impact entrainment rather than fluid entrainment.

The existing results showed that the saturation time of wind-blown sand is about 2 seconds. Therefore, we did not provide any discussions about the time evolution. We also did not analyze the results of 0.5 seconds, but focused on the effect of SCBs on the wind-blown sand within 2 to 5 seconds after the wind-blown sand is saturated. Thank you for the reminder, and we have removed the result of 0.5 seconds in the revised manuscript.

[Figure]

**Figure 13.** The top view of the particle positions of the wind-blown sand in presence of SCBs at the time $t$=2.0 s (a) and $t$=5.0 s (b), where U represents the speed of the particles ($u_\tau$=0.3 m/s, N=5~20 m). The y coordinates are correspondingly shifted up by 7.5 per case.

[Figure]

**Figure 14.** The top view of the particle positions of the wind-blown sand in presence of SCBs at the time $t=2.0$ s (a) and $t=5.0$ s (b), where U represents the speed of the particles ($u_\tau=0.6$ m/s, N=5~20 m). The y coordinates are correspondingly shifted up by 7.5 per case.

**Finally,** thank you for your help. The authors have accepted Referee2's advice, and modified the paper carefully in the revised version. We hope our efforts can make a little progress in this paper.

---

## Author Comment (AC2)

Authors' Responses to the Comments on the Manuscript

**"Modeling the Inhibition Effect of Straw Checkerboard Barriers on Wind-blown Sand"**

**General Response to the Comments and the Suggestions:**

According to the suggestion of the **Referee2**' comments, we give a substantial revision of the original manuscript such that a clear description of the research is displayed in the revised version. Many Thanks **Referee2** for reviewing of the manuscript carefully and objectively, and the comment: '**interesting**', '**positive significance**', '**important contribution**', which is a great affirmation to author's work. We have carefully revised the language and format of the manuscript mentioned by **Referee2**, and the detailed response are as follows.

**Responses to Comments of Reviewer#2:**

**General comments:**

*This is an interesting paper that uses numerical simulation to study the inhibition effect of straw checkboard barriers (SCBs) on wind-blown sand, and also the influence of SCB's laying length was discussed. Based on the simulation results, the wind field, particle concentration and transport rules around the SCBs are revealed and analyzed. These works have positive significance for people to deeply understand the function of SCB and effectively improve its use effect. This article thus has the potential to be an important contribution. However, there are several major issues with the article.*

**Item 1:**

*First of all, it is suggested that the language and format of the full text should be carefully examined and revised. There are many obvious grammatical and formatting errors, such as in line 95-96, line 190, 195, 198, 220, 226, line 231-232, line 273, line 314-318…*

**Authors' Response 1:**

Thank Referee for the reminder.

The authors have modified the original manuscript carefully on the grammar and formatting errors, and we have made many changes in the revised version, please see lines 94-97, 137, 191, 196, 199, 221, 227, 232-233, 253, 275-277, 316-320 for detailed information.

For example:

"Moreover, since the three-dimensional SCB is simplified into a two-dimensional plane with only the streamwise direction and vertical direction, the impact of this simplification is uncertain."

"where $i = 1, 2$ and 3 correspond to the streamwise,"

"where $k_n = 2 \times 10^6$ is the normal stiffness coefficient;"

"where $m_i$ and $m_j$ are the mass of particle $i$ and $j$,"

"where $k_t = 2 \times 10^6$ is the tangential stiffness coefficient,"

"where $n_0$=0.4, $A$=0.68, $B$=0.39, $\zeta$=5, $C$=0.92, and $D$=1.39 (Huang et al., 2017)."

"where $v_{ej}$ is the ejection speed and the overbar represents a mean value"

"Following the idea of Dupont et al. (2013), aerodynamic entrainment is not considered in our model."

"where $C_d$ is the drag coefficient,"

"We first simulate the clean air field flow in the presence of SCBs for 30 seconds to obtain the full development of the flow field. Then, we add sand particles to the flow field to obtain a sand-laden flow."

"This is because the streamwise position of $x = 6$ m is in the peak region of the "overshoot" phenomenon, while the streamwise positions of $x = 1$ m and $x = 14.5$ m are in the rising region and stable region, respectively. Due to the massive accumulation of sand particles near the surface (0-20 mm), the concentrations cannot be easily measured."

We also had the article retouched using some proof-reading services. Perhaps it can improve our writing ability as soon as possible.

**Item 2:**

*Line 161, In Figure 1 we don't see any information about the inlet condition setting.*

**Authors' Response 2:**

Thank Referee for the reminder.

We should not have misled the reviewer by giving a figure number of the reference. The authors have corrected it. Please see lines 160-163 in the revised manuscript.

", the LWS method (Lund et al., 1998) is applied to the inlet condition and the recycling plane at $x_{ref}/Lx$=12.5% (Inoue and Pullin, 2011). $x_{ref}$ =5 m is the position of the recycling plane, and $Lx$=40 m is the total length of the flow direction."

**Item 3:**

*Line 254-256, is there any evidence to confirm that SCB can be approximated as vegetation when evaluate rfa in equation 16?*

**Authors' Response 3:**

Thank Referee for the question.

This resistance force equation is widely used to describe the equivalent resistance of vegetation, shub, tree and so on (Wilson, 1988, Li et al., 1990, Green, 1992, Katul et al., 2004, Dupont et al., 2014). In recent years, this formula has also been extended to describe the equivalent resistance of SCBs (Bo et al., 2015).

Here is the reference:

Bo, T. L., Ma, P., and Zheng, X. J.: Numerical study on the effect of semi-buried straw checkerboard sand barriers belt on the wind speed, Aeolian Res. 16, 101-107, 2015.

**Item 4:**

*How to describe the dynamic behavior after the collision of saltation particles and SCB?*

**Authors' Response 4:**

It a good question. Because of the limitation of the drag force method, the SCBs only affect the velocity of the flow field rather than the real presence. Our model considers more the effect of the flow field on the particles and does not simulate the collision process between the particles and the SCBs.

In the section "Conclusions and outlook" of the original manuscript, the limitations of our model are described:

"Although our model has been able to reveal the inhibition effect of the SCBs on wind-blown sand, there are still some aspects that need improvement, such as aerodynamic entrainment, particle deposition on the SCB, and collision between the sand particles and the SCBs."

**Item 5:**

*Line 264, what is wall-normal direction. There are several walls in the simulation region.*

**Authors' Response 5:**

When defining the flow field, we emphasized that the $z$ direction is wall-normal direction. In our model, the inlet, outlet, spanwise boundary and upper boundary are not wall conditions, but only the bottom surface is wall condition. Therefore, wall-normal direction is always used to represent the vertical direction in this paper.

$$\frac{\partial \tilde{u}_i}{\partial t} + \tilde{u}_j \frac{\partial \tilde{u}_i}{\partial x_j} = -\frac{1}{\overline{\rho}_f} \frac{\partial}{\partial x_i} (\tilde{p} - \nu \frac{\partial \overline{\rho}_f \tilde{u}_j}{\partial x_j}) - \frac{\partial \tau_{ij}}{\partial x_j} - \delta_{i3} g (\frac{\tilde{\theta}}{\overline{\overline{\theta}}} - \frac{c_p}{c_v} \frac{\tilde{p}}{\overline{p}}) + \frac{F_i}{\overline{\rho}_f},$$

where $i = 1$, 2 and 3 correspond to the streamwise, spanwise and wall-normal directions (i.e., $x_1 = x$, $x_2 = y$, $x_3 = z$, $u_1 = u$, $u_2 = v$, $u_3 = w$), respectively.

**Item 6:**

*Line 293, what are delta t, H and M in Eq. 17? What is the meaning of 'mass in the range'? Is it similar to concentration? Why dx is divided here? From the physical concept, the scale information in the x direction should not appear here (should be y direction). Anyway, please check and define all the variables involved in the Eq. 17 and give the dimension of q.*

**Authors' Response 6:**

Thank you for your reminder.

The streamwise sand transport rate means the mass of sand particles passing through the plane perpendicular to the flow direction in a unit time and unit length. What we calculate is the sand transport rate in each interval of the flow direction. The unit of $q$ is kg/m/s.

We have modified the Eqs. (17) and (18).

The original:

$$q = \sum_{z=0}^{z=H} \sum_{y=0}^{y=M} m(x) / \Delta x / \Delta t_s.$$

$$C = \sum_{z=0}^{z=H} \sum_{y=0}^{y=M} \sum_{x=0}^{x=L} m(x) / Lx / Ly / Lz.$$

The revised:

$$q(x) = \sum_{z=0}^{z=Lx} \sum_{y=0}^{y=Ly} m(x, y, z) / \Delta x / \Delta t_s.$$

$$C = \sum_{z=0}^{z=Lz} \sum_{y=0}^{y=Ly} \sum_{x=0}^{x=Lx} m(x, y, z) / Lx / Ly / Lz.$$

**Item 7:**

*Line 309, what is the difference between the transport rate density and the transport rate defined in eq. 17?*

**Authors' Response 7:**

Thank you for your question.

The integral of sand transport rate density along the height is the sand transport rate.

$$q(x) = \int_{0}^{Lz} q(x, z) dz$$

**Item 8:**

*Line 301-350, the author spent a great deal of space to analyze and discuss the structural characteristics of aeolian sand flow without SCBs, but it seems that this is not the focus of this paper. Appropriate reduction is recommended.*

**Authors' Response 8:**

Thank you for your suggestion.

These contents are the verification part of our code, which is very important. In our previous article (*Huang, H. J.: Modeling the effect of saltation on surface layer turbulence, Earth Surf. Proc. Land. 45(15), 3943-3954, 2020.*), we verified the wind speed and speed fluctuations of clean air flow, but did not discuss the structural characteristics of aeolian streamers. In this manuscript, we want to add these contents to the verification part and compare it with the experimental results of Baas and Sherman (2005). The whole content is a good complement to the part of the program verification. We use the simulated aeolian streamers to compare with the existing results, which will help readers to recognize our results more. In addition, the streaks of the clean air flow in presence of SCBs have been shown in the results and discussion section, which can let the readers intuitively feel the difference.

**Item 9:**

*In the part of model validation, the verification of the simulation results of sand flow with SCBs is not sufficient. The qualitative comparison cannot prove that the simulation results of the adopted model are credible in the presence of SCBs. Some quantitative comparisons are necessary. I believe the author should be able to find the observation data of the sand flow in the presence of SCB.*

**Authors' Response 9:**

Thank you for your suggestion.

There are few directly comparable experimental data. We found one experimental work that is close to our SCBs model. According to the suggestion of reviewer, we made a comparison from the following aspects.

In the verification part, we supplement the simulation results of the velocity profile with SCBs and compare them with the experimental results. The experimental data we used are from Tao et al. (2020). Accordingly, we have added Figure 6b to the revised manuscript and illustrated it. Please see lines 390-403 in the revised manuscript.

[Figure]

**Figure 6. (b)** The dimensionless wind speed varying with dimensionless height, comparison between our simulated results and the existing experiment.

"In addition, we compared the wind speed profiles with the experimental results of Tao et al. (2020). In Figure 6b, the streamwise wind speed in the horizontal coordinate is dimensionless with the reference wind speed $U_{ref}$ and the height in the vertical coordinate is dimensionless with the reference height $z_{ref}$. In the wind tunnel experiment conducted by Tao et al. (2020), the maximum boundary layer thickness is given as 0.5 m, so the reference height is taken as $z_{ref}$=0.5 m. Then, the wind speed at $z$=0.5 m is determined as the reference wind speed $U_{ref}$=12.58 m/s based on their inlet wind profiles. In our simulation case, $z_{ref}$=0.4 m is the initial inlet boundary layer thickness, and $U_{ref}$=16.3 m/s is the reference wind speed. We select the experimental results of wind speed profiles at the SCB belt positions $x_0$=2.2 m and 2.8 m along the flow direction to compare with our numerical results at $x_0$=2.5 m (streamwise position $x$=7.5 m). The dimensionless results show that our results are consistent with the experimental results in quantitative and qualitative, which indicates that our model can well reveal the inhibition effect of SCBs on the flow field."

Here are the references:

Wang, T., Qu, J. J., and Niu, Q. H.: Comparative study of the shelter efficacy of straw checkerboard barriers and rocky checkerboard barriers in a wind tunnel, Aeolian Res. 43, 100575, 2020.

**Item 10:**

*Line 434-443, the sand accumulation pattern in a single SCB should be related to the vortex structure of the local flow. It seems to be too far-fetched to explain it only from the result of time-averaged wind speed.*

**Authors' Response 10:**

Thank you for your question.

Our SCBs model is based on the volume resistance force method, which is different from the SCBs completely equivalent to the solid wall. If the SCB is a solid

wall, there will be obvious vortex structure similar to that in the backward-facing step flow, such as Xu et al. (2018, JGR)

[Figure]

**Figure 4.** Contour maps of velocity in the center section. (a) Mean horizontal velocity;

However, this backflow vortex phenomenon does not appear in our model, because the wind is able to pass through the SCBs (only weakened, not very strong backflow), which is also closer to reality. In addition, the side length of our SCBs is 1 m, which is twice their side length (0.5m), so this regular backflow vortex phenomenon is not significant in our model.

[Figure]

Figure. The side view of X-Z plane streamwise velocity after containing the SCBs ($u_\tau$=0.6 m/s, N=5~20 m, $y$=0 m).

What's more, we have zoomed in on the speed diagram inside the SCBs, and you can see:

The streamwise speed in most SCBs is higher in the central area than in the surrounding area (red box), and the streamwise speed in a few SCBs is higher in the surrounding area than in the central area (yellow box).

[Figure]

[Figure]

This conclusion is also quite consistent with the actual situation. The wind speed in the center of a single SCB is high, and then the sand particles are deposited less; the wind speed around the surrounding area is small and the sand particles are deposited more. Such accumulation phenomenon is also consistent with the experimental results (as shown in the figure above). Therefore, we believe that the corresponding deposition pattern can be explained by the time-averaged wind speed. I hope that it will be approved by the reviewer.

**Item 11:**

*Line 466, why does the laying length of SCBs affect the sand transport rate in the upwind area (x<5m)?*

**Authors' Response 11:**

Thank you for your question.

The exit and entrance boundaries of the particles motion are set as periodic boundaries. Our initial particles are randomly distributed, and no more particles will be added later. Therefore, the particle information at the entrance of the whole wind-blown sand flow depends on the particle information at the exit, which can ensure that there are particles at the entrance of the whole wind-blown sand flow all the time. It does not affect the steady-state results because the sand transport rate in the wind-sand flow before the SCBs entrance has decreased to a consistent level.

[Figure]

**Figure 12.** The streamwise sand transport rate in the different laying length cases ($u_\tau$=0.6 m/s, N=5~10 m, 5~20 m, 5~30 m). Dark lines are the result of smoothing.

**Item 12:**

*Line 474, 478-479, it is a little strange here that the author did not consider the fluid entrainment. How does the wind-blown sand flow recover in the downwind area of the SCBs where the sand transport rate has reduced to zero?*

**Authors' Response 12:**

Thank you for your question.

The sand transport rate you mentioned is close to 0 in some areas, but not exactly equal to 0. For example, in the laying length of N=5-30 m case, we draw the sketch map of the transient particle position inside the SCBs in the following figure. You can clearly see that the number of particles in the 20-30 m flow direction area is very sparse, but it has not completely disappeared. In order to show clearly, we have enlarged the particle size, but the actual particle size is very small. That is to say, as long as a few particles can pass through the SCBs area (in the statistical results, the sand transport rate of this part is infinitely close to 0), and then a stable wind-blown sand flow will soon develop after that.

[Figure]

(Original, N=5-30 m)

(Enlarged, N=5-30 m)

Here we want to explain to the reviewer that although fluid entrainment is very important, the existing literature showed that the minimum friction wind speed to maintain the wind-blown sand flow is the impact threshold rather than the fluid threshold. In other words, as long as there exists particle impact and the wind speed is above the impact threshold, the wind-blown sand flow can be continued. I hope our explanations can answer the doubts of reviewer.

**Item 13:**

*Line 481-482, is it possible that the length of the computation domain is not enough?*

**Authors' Response 13:**

Thank you for your question.

We think the computation domain is sufficient. From our wind field results, the mean wind speed tends to be stable, and the mean wind speed after adding the particles is also stable. The reason why the sand transport rate decreases is that the longer the SCBs are, the greater the fluid consumption is. Even if the wind speed behind the SCBs can recover, it is impossible to recover to the level of the inlet wind

speed, so the sand transport rate will decrease.

**Item 14:**

*The results in Figure 14 and Figure 12 do not seem to agree. Fig. 12 shows that there is almost no sand flow in the area of SCB, but Fig. 14 shows a different result.*

**Authors' Response 14:**

Thank you for your question.

This issue is consistent with **Item12**. The sand transport rate is statistically very low at N=5-20 m laying length case, but as long as some particles can pass through the SCBs, then under the condition that the wind speed behind the SCBs recovers to a certain level (larger than impact threshold), this part of particles can rapidly develop into wind-blown sand flow.

[Figure]

**Figure 14.** The top view of the particle positions of the wind-blown sand in presence of SCBs at the time $t$=2.0 s (a) and $t$=5.0 s (b), where U represents the speed of the particles ($u_\tau$=0.6 m/s, N=5~20 m). The y coordinates are correspondingly shifted up by 7.5 per case.

[Figure]

(Enlarged, N=5-20 m)

**Finally**, we want to say that we attach great importance to the Referee's comments, and take it seriously. Although there are still some aspects that need improvement, we are really making our contributions to the prevention of desertification.

**Once again, we thank the Referee's help.**